# Executive Functions of Adults with Binge-Eating Disorder: The Role of Weight Status and Psychopathology

**DOI:** 10.3390/brainsci12010006

**Published:** 2021-12-22

**Authors:** Nele Busch, Ricarda Schmidt, Anja Hilbert

**Affiliations:** Integrated Research and Treatment Center Adiposity Diseases, Behavioral Medicine Research Unit, Department of Psychosomatic Medicine and Psychotherapy, University of Leipzig Medical Center, Philipp-Rosenthal-Strasse 55, 04103 Leipzig, Germany; nele.busch@medizin.uni-leipzig.de (N.B.); ricarda.schmidt@medizin.uni-leipzig.de (R.S.)

**Keywords:** binge-eating disorder, obesity, executive functions, depression, psychopathology

## Abstract

Findings on executive functions (EFs) in binge-eating disorder (BED) are inconsistent and possibly biased by associated comorbidities. This study aimed to identify whether distinct levels of physical and mental comorbidity are related to EFs in BED. General and food-specific EFs in *n* = 77 adults with BED were compared to population-based norms and associations with weight status, depressive symptoms, and eating disorder psychopathology were analyzed. To detect within-sample patterns of EF performance, *k*-means clustering was applied. The results indicated that participants’ general EFs were within the average range with slight deficits in alertness. While depression and eating disorder psychopathology were unrelated to EFs, weight status was associated with food-specific attentional bias that was significantly higher in obesity class 2 than in overweight/obesity class 1 and obesity class 3. Four meaningful clusters with distinct strengths and impairments in general and food-specific EFs but without differences in clinical variables were identified. Altogether, adults with BED showed few specific deficits compared to normative data. Performance was unrelated to depression and eating disorder psychopathology, while weight status was associated with food-specific EFs only. The results highlight the need for longitudinal studies to evaluate the relevance of EFs in BED development and maintenance in neurologically healthy adults.

## 1. Introduction

Binge-eating disorder (BED) [1], characterized by recurrent objectively large binge-eating episodes without regularly occurring compensatory behaviors to prevent weight gain, is the most common eating disorder, with a lifetime prevalence of 0.9% to 1.9% in adults [2,3]. Physical and mental comorbidities are highly prevalent in BED, causing significant disease burden [4]. Particularly, 68% to 81% of patients with BED exceed a body mass index (BMI, kg/m²) of 25 kg/m² indicating overweight and obesity [3] and 65.5% have a major depressive disorder in their lifetime [5]. Although an individual’s neuropsychological profile based on specific deficits in executive functions (EFs) has been considered as a risk factor for the development and maintenance of BED [6], findings on neuropsychological characteristics in BED remain controversial.

EFs are cognitive proficiencies needed for complex intentional actions, with inhibition, working memory, and cognitive flexibility constituting the main components that build the basis for higher-order mechanisms such as reasoning, problem solving, and planning [7]. While some studies identified deficits in inhibitory control, working memory, planning, decision making, and problem solving in adults with BED compared to controls with overweight or obesity without BED [8,9,10,11], recent meta-analyses quantified deficient overall EFs in BED to be of small effect size only and evidence on impairment in specific EF subdomains remains inconclusive [12,13,14]. According to Kittel et al., EF impairments in BED may be more pronounced in a food-specific than general context, but evidence is still low in this respect [15]. For example, women with BED showed memory bias in the context of food stimuli in a recent-probes task [16] and adolescents with BED detected food stimuli faster than neutral stimuli in a visual search task, indicating an attentional bias towards food [17]. Importantly, data on EFs in adults with BED are based on mostly small samples with BED, with the majority of studies including less than 35 participants with BED. Additionally, normative performance evaluations on computerized tasks are virtually lacking, which would involve comparisons with population norms. Thus far, there is only evidence indicating greater self-reported difficulties in EFs in individuals with obesity and BED compared to normative data [18]. A ranking of objective neurocognitive performance of BED within healthy controls was provided by Dingemans et al., who defined EF deficits in a sample of 91 adults with BED as one standard deviation below the mean of the control group (*n* = 56). Accordingly, 67% to 90% of participants with BED were classified within the healthy range in each computerized test administered in the study; however, the authors did not provide particular information on specific EF components (response inhibition, decision making, set shifting, working memory, and central coherence) [19].

Notably, studies on EFs in BED show a large variability in weight status with mean BMIs ranging from 25.6 kg/m² [20] to 45.4 kg/m² [21] and include individuals with and without mental comorbidities, making comparisons between studies difficult. In this context, it is necessary to scrutinize whether EF deficits in BED are attributable to BED-related comorbidities. For example, overweight was shown to be associated with deficits in inhibition and working memory compared to controls with normal weight. EF deficits in individuals with obesity were found to be even broader than in those with overweight—additionally affecting cognitive flexibility, decision making, verbal fluency, and planning [22]. Adults with major depressive disorder consistently showed reduced inhibition, cognitive flexibility, working memory, verbal fluency, and planning compared to healthy controls, independent of the presence of BED, with more depressive symptoms being related to more impaired EFs [23]. The only study systematically analyzing group differences in EFs of individuals with BED as a function of depressive symptoms (no to mild symptoms versus moderate to severe symptoms) did not find more impaired EFs with increasing level of depression in neurocognitive tasks, but in self-reported EFs in daily life [19]. Accordingly, a recent study in *n* = 66 adults with BED demonstrated significant correlations between self-reported depressive symptoms and lower cognitive flexibility, as measured by the Trail Making Test and the Wisconsin Card Sorting Task [24]. Regarding eating disorder psychopathology, with high levels being characteristic for BED [1,25], significant correlations were found with worse response inhibition as well as with higher attentional bias towards food in women with BED and weight-matched controls [10,26]. Similarly, more severe binge-eating pathology as measured by the Binge Eating Scale [27] was negatively correlated with decision making in a study of women with BED and controls without BED across the weight spectrum [28], suggesting that eating disorder psychopathology may have a deteriorating influence on EFs in individuals with BED or vice versa.

Beyond these preliminary approaches to identify factors that may explain the het-erogeneity in findings on EFs in BED across studies, within-sample heterogeneity in EFs in individuals with BED has not been examined. A statistical approach to identify within-sample differences in EF performance is cluster analysis, which has been previously used to subtype adults with BED according to their similarities and differences in negative affect and dietary restraint [29,30,31] with the goal of adjusting treatment options to individual needs. Cluster analysis on EFs in eating disorders has only been conducted in a sample of *n* = 100 women with anorexia nervosa (AN) and Eating Disorders Not Otherwise Specified (EDNOS) [32]. Based on performance in set shifting, visuospatial constructional ability, and emotional theory of mind, three meaningful clusters emerged: a cluster of high to average functioning individuals, one with mixed performance showing only domain-specific EF deficits, and a third cluster with overall low performance. However, clusters did not differ in any of the examined demographic and clinical variables (e.g., age, illness duration, BMI, and eating disorder psychopathology).

In order to unravel how clinical characteristics that are connected to BED may be associated with EF deficits, the aim of this study was three-fold. First, we ranked the neurocognitive performance of *n* = 77 adults with BED within age-specific population norms, hypothesizing that this sample will perform worse than the community average. Second, we evaluated EF performance and measures of food bias as a function of weight status, depressive symptoms, and eating disorder psychopathology, hypothesizing that EF levels will gradually decrease with increasing weight status, depressive symptoms, and eating disorder psychopathology. Third, we aimed to identify underlying clusters of different EF performance profiles hypothesizing to find a cluster with average to high functioning, a cluster with deficits along the whole EF spectrum, and a cluster with distinct food-specific impairments, similar to evidence in AN and EDNOS [32]. As reported by Renwick et al., we hypothesized that clusters would not differ in age and sex, but expected the cluster with average to high functioning to have significantly lower weight status, level of depression, and eating disorder psychopathology than the other two clusters in accordance with our second hypothesis.

## 2. Materials and Methods

### 2.1. Participants and Procedure

Participants (*n* = 77) were enrolled within a randomized controlled trial on the efficacy of near-infrared spectroscopy neurofeedback for BED (NIRSBED; DRKS00014752, www.drks.de, accessed on 20 December 2021) and recruited through advertising and clinic referrals. For inclusion, age ≥ 18 years, a BMI ≥ 25 and < 45 kg/m², an interview-based diagnosis of BED according to the criteria of the Diagnostic and Statistical Manual of Mental Disorders, 5th edition (DSM-5) [1], including BED of low frequency and/or limited duration, sufficient German language skills, and written informed consent to participate in the study were required. Exclusion criteria comprised severe somatic or psychiatric conditions (e.g., psychosis, suicidality, substance use disorders, attention-deficit/hyperactivity disorder (ADHD), and developmental disorders), impaired (uncorrected) hearing, vision, or speaking, a history of or planned obesity surgery, medication that could influence weight or EFs, pregnancy or lactation, and current psychotherapy focusing on weight, body shape, or eating behavior.

Interested participants were screened by telephone and invited for an in-person appointment at the study center if eligible. BED was diagnosed using the BED module of the Eating Disorder Examination (EDE) [33,34], administered by trained doctoral candidates and student assistants who were supervised regularly. Included participants completed a battery of computerized neuropsychological tests assessing general EFs provided by the Vienna Test System by Schuhfried GmbH [35] and by Mueller et al. [36] as well as food-specific EFs by Millisecond Software LCC [37]. Data on EFs and clinical parameters were assessed before the beginning of the treatment. The study was approved by the Ethics Committee of the Medical Faculty of University of Leipzig (476/17-ek).

### 2.2. Measures

#### 2.2.1. Demographic and Clinical Parameters

Data on age, sex (male, female), and educational level (<12, ≥12 years) were assessed via self-report. BMI was computed based on body weight and height measured with calibrated instruments. Weight status was classified into overweight (25 ≤ BMI < 30 kg/m²), obesity class 1 (30 ≤ BMI < 35 kg/m²), obesity class 2 (35 ≤ BMI < 40 kg/m²), and obesity class 3 (BMI ≥ 40 kg/m²) according to the World Health Organization [38] (p. 557).

The level of depression was measured using the nine-item depression module of the Patient Health Questionnaire (PHQ-9) [39,40]. Participants were asked how often specific symptoms, for example, lack of concentration or suicidality, interfered with their daily life. Each item was rated on a four-point rating scale (0 = not at all, 3 = almost every day). A sum score was calculated and evaluated categorically as level of depression (0–4 = no/minimal depression, 5–9 = mild, 10–14 = moderate, 15–19 = moderately severe, 20–27 = severe [41]); Cronbach’s alpha was 0.78 in this study.

Eating disorder psychopathology was measured by the Eating Disorder Examination-Questionnaire (EDE-Q) [42,43]. In 22 items, participants were asked to estimate the frequency or intensity of eating disorder-related thoughts and behaviors during the past 28 days on a seven-point rating scale (0 = no days/not at all, 6 = every day/extremely). The global mean score was calculated as an indicator for overall eating disorder psychopathology (Cronbach’s alpha in this study = 0.84). Participants were then classified depending on their percentile, as derived from age- and sex-specific normative data [44]: global score ≤ 95th percentile, 95th < global score < 99th percentile, and global score ≥ 99th percentile.

#### 2.2.2. Decision Making

The Cards and Lottery Task (CLT) [36], a newly developed computerized card game, was used to assess short- and long-term decision making. In each of the 36 trials, participants were asked to draw a card from one of two decks with different cards depicting certain virtual amounts of money (−150 to 250 EUR) that were then added to or, in case of negative values, subtracted from a virtual account balance. The overall aim was to gain as much virtual money as possible. In addition to those short-term consequences, a virtual jackpot of 5000 EUR was added or subtracted at the end of the game. The chance of winning this jackpot could be influenced by the choices the participants made during the game: apart from a particular amount of money, some cards showed either a star or bomb symbol, with stars increasing the probability of winning the lottery and bombs deteriorating the probability of winning the lottery. For every participant, the number of advantageous decisions (NAD) reaching from 0 to 36 was calculated with lower scores representing riskier decision making that focuses on positive short-term consequences, despite future negative effects [36,45].

#### 2.2.3. Response Inhibition

In a Stop-Signal Task (SST) [46], participants were presented 200 arrows that either pointed to the left or to the right with the instruction to press the 5- or 6-key as quickly as possible depending on the direction of the arrow. A total of 48 trials were randomly followed by a stop signal, a 1000 Hz sound, indicating that participants must withhold button press. The interval between the arrow and the stop signal (stop-signal delay, SSD) was adjusted to participants’ performance. The difference between the mean reaction time (RT) and the mean SSD was calculated as stop-signal reaction time (SSRT) with higher SSRTs representing impaired response inhibition.

A Go/No-Go Task (GNG) [46] was used to test the ability to restrain an action that was not yet initiated. In 250 trials, participants were presented either a triangle or a circle on a computer screen. They were instructed to react to triangles by pressing the right shift key and to withhold their reaction when a circle appeared on the screen. Failures to do so were counted as commission errors, indicating impaired response inhibition. The Go/No-Go ratio was 81% vs. 19%.

#### 2.2.4. Working Memory

Working memory was examined with the N-Back Verbal Task (NBV) [47] that comprised 100 consonants being randomly presented for 1.5 s, each followed by an interstimulus interval of 1.5 s. Participants were asked to press the button if the currently shown letter was equal to the letter shown two trials ago. Higher numbers of correct responses represent better working memory.

#### 2.2.5. Cognitive Flexibility

Cognitive flexibility was assessed using part B of the computerized version of the Trail Making Test (TMT) [48]. Participants were presented the numbers 1 through 13 and the letters A through L along with the instruction to alternately click on numbers and letters in ascending or alphabetical order with a computer mouse. The time it took participants to solve this task serves as an indicator for cognitive flexibility with shorter completion times representing higher levels thereof.

#### 2.2.6. Planning Ability

Planning ability was examined using a computerized form of the Tower of London (ToL) [49]. Participants were presented a wooden rack with three poles and three differently colored balls stacked on these poles in various start constellations. In each of the 28 trials, participants were asked to reach a certain goal arrangement by moving the balls with a computer mouse, minding that the balls were only free to be moved if they were in the top position and that each pole could only hold a maximum number of balls depending on its height. There were different levels of difficulty, determined by the minimum number of moves that were required to solve the task, ranging from 3 to 6. The number of solved 4- to 6-move trials was counted as an indicator of better planning ability.

#### 2.2.7. Alertness

Alertness, i.e., the ability to be sensitive and prepared for arising stimuli [50] was assessed using a unimodal subtest of the “Perception and Attention Functions Battery Alertness” (WAFA) [51]. In this subtest, participants were asked to react to 25 black circles that were presented for 1500 ms with an interstimulus-interval of 3 to 5 s as quickly as possible. Higher logarithmic mean RTs indicate lower levels of alertness. 

#### 2.2.8. Attentional Bias to Food Cues

A food-specific Dot Probe Task (DPT) [52] was used to examine a potential attentional bias towards food cues. In 80 trials, participants were presented a fixation cross for 500 ms on a computer screen, followed by two pictures on the left and right part of the screen—one depicting a neutral object and the other one depicting food in 40 of the trials. Two pictures of neutral objects were shown in the other 40 trials serving as fillers that were not used for calculation of attentional bias towards food. After 1000 ms, one of the two pictures was randomly exchanged by an “X” as probe. Participants were instructed to react as quickly as possible by pressing the E- or I-key, depending on the position of the probe. Latency times were recorded for each trial. The difference between mean RTs to probes replacing neutral pictures and probes replacing food pictures was calculated for each participant as an attentional bias score [53]. Positive scores indicate an attentional bias towards food cues.

#### 2.2.9. Approach Bias to Food Cues

A food-specific Approach Avoidance Task (AAT) [54] was used to examine a potential approach bias towards food cues. In each of the 80 trials, participants were presented a picture of one of the following categories: high-caloric food, low-caloric food, appetitive neutral (e.g., leisure gadgets), and boring neutral (tools and office utensils). Participants were instructed to react as quickly as possible by pushing a joystick away from them if a picture was presented in landscape format and by pulling the joystick towards them if the picture was in portrait format. Latency times were recorded for each trial. The median RT for pulling-trials was subtracted from the median RT for pushing-trials in each category to calculate a specific difference score. A positive output indicates an approach bias for the respective category. In this study, only the approach bias towards high-caloric food was used.

### 2.3. Statistical Analyses

To compare EF performance of individuals with BED to the population, age-specific *T* scores for tasks from the Vienna Test System (SST, GNG, NBV, TMT, ToL, WAFA) were derived from population-based norms provided by Schuhfried GmbH (*n* = 356–419) [35]. *T* scores for the CLT were calculated using published data of *n* = 70 adults with no current or past neurological or mental diseases [36]. For tasks of general EFs, participants with *T* scores ± 3 *SD* of the sample mean were individually evaluated for plausibility and discarded as outliers if necessary, resulting in at most *n* = 1 outlier per test. Outliers for the food-specific AAT were determined according to Wiers et al. [54], excluding participants with errors on more than 25% of the trials and mean RTs longer than 3 *SD* from the sample mean (*n* = 9 participants). We applied the same rules to the DPT, excluding *n* = 1 participant. Due to technical reasons and early task cancellation, outcomes were missing for *n* = 6 (SST, AAT), *n* = 5 (CLT, ToL), *n* = 3 (DPT), and *n* = 1 (GNG, NBV, TMT, WAFA). Regarding EDE-Q and PHQ-9, data were missing for *n* = 3 participants. Appendix A provide an overview of the number of participants included in analyses on general and food-specific EFs.

To test for possible differences between participants with higher vs. lower level of overweight, depression, and eating disorder psychopathology in general EFs, we computed multivariate analyses of variance (MANOVAs) with weight status (25 ≤ BMI < 35 kg/m², 35 ≤ BMI < 40 kg/m², BMI ≥ 40 kg/m²), level of depression (PHQ-9 score < 5, 5 ≤ PHQ-9 score < 10, 10 ≤ PHQ-9 score ≤ 27), and level of eating disorder psychopathology (EDE-Q ≤ 95th percentile, 95 < EDE-Q < 99th percentile, EDE-Q ≥ 99th percentile) as categorical, independent variables and measures of general EFs as dependent variables. The same analytic steps were conducted with measures of food bias as dependent variables. An a priori power analysis using G* Power Version 3.1 [55] revealed a sample size of *n* = 69 was needed to detect medium effects (*f*^2^ = 0.15) [56] (pp. 413–414) with a power of 1 − β = 0.80 for a MANOVA including seven general EF measures. Although our original sample size of *n* = 77 would have sufficed, MANOVAs were only calculated for *n* = 57–60 due to missing data and outliers, resulting in a post-hoc power of at least 1 − β = 0.70. For the two food bias measures, the needed sample size of *n* = 45 was exceeded to detect medium-sized group differences in MANOVAs. Due to the high rates of missing data, MANOVAs were repeated with inclusion of outliers and exclusion of tests with ≥ 3 missing data in sensitivity analyses, which were only reported if they changed the main results. Significant multivariate results were followed up by ANOVAs for each neurocognitive task separately. In case of violation of homoscedasticity, ANOVAs were followed up with Kruskal–Wallis one-way analyses of variance, which were only reported if they changed the results. Multivariate η^2^ and partial η^2^ were used to estimate effect sizes for MANOVAs and ANOVAs with 0.01 ≤ η^2^ or η_p_^2^ < 0.06 representing small, 0.06 ≤ η^2^ or η_p_^2^ < 0.14 representing medium, and η^2^ or η_p_^2^ ≥ 0.14 representing large effects [56] (pp. 284–287). Exploratively, categorical analyses were supplemented by simple linear regression analyses between weight status, level of depression, and eating disorder psychopathology as predictors, and measures of EFs as criteria using continuous data. The results will be reported in the Appendix A (see Appendix A).

A *k*-means cluster analysis on general and food-specific EF measures was performed with the aim to detect distinct patterns of EF performance within the sample. An appropriate number of clusters was detected through comparisons of two- to six-cluster solutions regarding adequate cluster sizes, impact of single variables on cluster discrimination, parsimony, and interpretability. To ensure comparability, raw data of food bias were first *z* standardized, then reversed so that lower *z* scores indicate higher food bias, and transformed into *T* scores. The above-mentioned *T* scores were used for the general EF measures. To further investigate possible associations of demographic (age, sex, educational level) as well as clinical variables (BMI, EDE-Q global score, and PHQ-9 score) with cluster membership, Pearson’s chi-squared test, Fisher-Freeman-Halton exact test, and one-way ANOVAs were conducted. All statistical analyses were performed using IBM SPSS Statistics Version 25.

## 3. Results

### 3.1. Participants

The majority of the total sample (*n* = 77) were women (*n* = 60, 78%). Mean age was 46 years, ranging between 21 and 76 years. Dichotomous categorization of the educational level (<12, ≥12 years) yielded two almost equally sized groups of lower (*n* = 40, 52%) and higher education (*n* = 37, 48%). Participants’ mean BMI was 37 kg/m², with 38% (*n* = 29) having obesity class 3, 26% (*n* = 20) each having obesity class 1 and 2, and 10% (*n* = 8) being overweight. The mean EDE-Q global score was 2.84 (*SD* 1.08), with 32% (*n* = 24) having a global score ≤ 95th percentile, 41% (*n* = 30) scoring ≥ 95th but < 99th percentile, and 27% (*n* = 20) exceeding the 99th percentile. The mean PHQ-9 score was 8.71 (*SD* 4.49) and both moderate to severe (10 ≤ PHQ-9 ≤ 27) as well as mild depression (5 ≥ PHQ-9 < 10) were each present in 41% (*n* = 30) of the sample, whereas 19% (*n* = 14) of the sample had no to minimal depression (PHQ-9 < 5).

### 3.2. Neurocognitive Measures

#### 3.2.1. EF Performance Compared to Population Norms

*T* scores for EF performance of the total sample based on community norms are depicted in Table 1. For most of the measures, the mean *T* score was below 50 but still over 40, demonstrating average performance. The proportion of the sample with below average performance ranged from 10.7% to 52.0% across tasks, while 0.0% to 15.8% performed above normative average. Lowest mean EF normative scores were obtained for alertness (WAFA; *M* = 39.2) and decision making (CLT; *M* = 40.3), indicating slight deficits relative to the normal population.

#### 3.2.2. Group Differences in EF Performance

MANOVAs on general EFs did not reveal any significant associations with participants’ weight status, *F*(14, 102) = 1.198, *p* = 0.289, η_p_^2^ = 0.14, level of depression, *F*(14, 96) = 1.197, *p* = 0.290, η_p_^2^ = 0.15, and eating disorder psychopathology, *F*(14, 96) = 1.170, *p* = 0.311, η_p_^2^ = 0.15. As depicted in Table 2, the MANOVA on food bias measures as dependent variables revealed significant effects for weight status, indicating that weight status was significantly associated with attentional bias towards food as measured by the DPT, without significant effects on approach bias to high-caloric food. Bonferroni-corrected post-hoc comparisons disclosed that attentional bias for food was significantly higher in those with obesity class 2 than in those with overweight and obesity class 1, and in those with obesity class 3 that did not differ significantly from each other (*p* > 0.05). No significant effects on measures of food bias were found for level of depression, *F*(4, 110) = 0.305, *p* = 0.874, η_p_^2^ = 0.01, and eating disorder psychopathology, *F*(4, 110) = 1.681, *p* = 0.159, η_p_^2^ = 0.06. Including sex and educational level as covariates did not change the results.

### 3.3. K-Means Cluster Analysis

A four-cluster solution fit the data on EF performance best, yielding four differently sized clusters: Cluster 1 (*n* = 29), Cluster 2 (*n* = 12), Cluster 3 (*n* = 21), and Cluster 4 (*n* = 15). As shown in Table 3, clusters differed significantly in all of the included variables except for response inhibition (GNG) and alertness (WAFA) with large effect sizes. Cluster 1 showed low to average general EFs, specifically, deficits in decision making (CLT) and alertness (WAFA), without any food-specific deficits. This cluster was, therefore, named the “low general EFs” cluster. Cluster 2 constituted the “low general and food-biased EFs” cluster, since EF performance was similar to Cluster 1, supplemented by substantial food bias. Both Cluster 3 and Cluster 4 were characterized by average to high general and food-specific EFs aside from low levels of alertness (WAFA). Because Cluster 3 showed specific strengths in planning (ToL) and cognitive flexibility (TMT), it was labeled as the “high EFs” cluster, while Cluster 4 was characterized as the “average EFs” cluster. As depicted in Table 4, clusters did not differ in age and sex, but Clusters 3 and 4 included a significantly greater proportion of participants with higher education than expected, while Clusters 1 and 2 included significantly smaller proportions of participants with higher education than expected. No significant differences between clusters were seen in the clinical variables of BMI, level of depression, and eating disorder psychopathology (*p* > 0.05).

## 4. Discussion

The aim of this study was to identify factors explaining between- and within-sample differences in EF performance in adults with BED by evaluating the effects of underlying clinical characteristics including weight status, depression, and eating disorder psychopathology on EFs. Contrary to recent etiological theories for BED [6], this study revealed only minimal deficits in EF performance in an adult sample of *n* = 77 treatment-seeking participants with BED compared to community-based norms, although a substantial proportion of adults with BED (11–52%) performed below normative average in the EF domains that were assessed in this study. BMI, level of depression, and eating disorder psychopathology did not significantly determine the extent of EF impairment with an exception for weight status being significantly associated with levels of attentional bias towards food. To detect patterns of EFs within the sample, *k*-means cluster analysis was used, yielding four clusters that differed in neurocognitive test performance, but not in demographical and clinical characteristics, except for education.

Concerning our first hypothesis, the present sample scored within the average range for most measures of the general EFs compared to normative data, except for alertness, which had a mean *T* score of 39, indicating slight deficits in this EF component. Notably, ≥20% of participants performed below normative average in tasks assessing decision making, alertness, planning, and response inhibition, supporting the assumption that EFs are not generally impaired in all individuals with BED, but rather in a certain proportion, which may account for small-sized overall EF deficits in meta-analytic studies [13,14]. Importantly, since no normative data were available for tests using food stimuli, conclusions about participants’ food-specific EFs cannot be drawn. In this context, previous meta-analyses [13,14] explicitly excluded food-specific EFs from their analyses, so that evidence on these specific EFs is still based on single studies with varying results [15,16,17,57,58,59]. Given the variety in tasks used to assess food-specific EFs in the literature, standardized food-specific measures are urgently needed—ideally with population-based norms in order to evaluate the relevance of food-specific EFs for adult BED. 

Contrary to our second hypothesis, weight status only explained group differences in food-specific EFs, while no differences in general EF performance were obtained, al-though large effects were demonstrated. Specifically, participants with obesity class 2 showed the highest attentional bias to food without significant differences between those with lower (overweight, obesity class 1) and higher (obesity class 3) weight status with a medium to large effect. These findings indicate that, unexpectedly, there may be no linear relation between weight status and food attentional bias. Non-linear associations with BMI have been previously reported for reward sensitivity [60], food addiction symptomatology [61], cognitive restraint towards food [62], dopamine receptor binding potential [63], and brain activity during regulation of food craving [64] in individuals from the community across the weight spectrum and candidates for obesity surgery. These factors may play a mediating role in the non-linear association between attentional bias towards food and BMI found in this study, warranting further research in individuals with BED.

Against hypotheses, the level of depressive symptoms and eating disorder psychopathology in adults with BED did not significantly account for variability in general EFs despite large-sized effects nor did they account for variability in food-specific EFs, revealing small-to-medium-sized effects. Similar results were obtained by Dingemans et al. [19], who found that adults with BED and moderate to severe depressive symptoms did not show significantly more impairments in computerized tasks compared to those with no to mild depressive symptoms and a healthy control group (BMI < 30 kg/m²). However, Dingemans et al. showed that deficits in questionnaire-based EFs in daily life were greater in individuals with BED with higher versus lower level of depression. Notably, neurocognitive tasks like those provided by the Vienna Test System were mainly developed in order to detect deficits in patients with objective brain damage [46,47,48,49,51] and might not be suited to reflect more fine-grained EF performance differences in neurologically healthy individuals. Subjective EF ratings may depict personal impairments associated with BED or may stem from a poor self-image, which is inherent to individuals with depression and BED [65,66]. The likely independence of psychopathology and EFs in BED is further reflected by the fact that lower levels of depression but not decision making, set shifting, working memory, and central coherence predicted higher chance of reducing eating disorder psychopathology and faster abstinence from binge eating in group cognitive behavioral therapy (CBT) for adults with BED [67].

Contrary to our third hypothesis that EF performance in BED would be mirrored by three different EF clusters as previously found in AN [32], specifically, a cluster with average to high functioning, a cluster with deficits along the whole EF spectrum, and a cluster with distinct food-specific impairments, a four-cluster solution fit the data best. Clusters differed significantly in seven of nine EF measures entered into the analysis with large effect sizes except for non-significant differences in response inhibition as measured by the GNG and in alertness measured by the WAFA. In line with the hypotheses, there was a highly performing cluster in general and food-specific EFs (Cluster 3) and an averagely performing cluster throughout all clustering variables (Cluster 4). However, contrasting expectations, there were two low to averagely performing EF clusters, with one of them showing only deficits in general EFs (Cluster 1) and the other cluster additionally presenting with food-specific impairments (Cluster 2). The results, thus, indicate that EF performance may be more fine-grained than in individuals with AN [32], although the number and types of EF measures used for clustering differed between the present and Renwick et al.’s study. Demographically, percentages of higher education were greater in the two higher functioning Clusters 3 and 4 than in the lower functioning Clusters 1 and 2, indicating that a higher educational level may be associated with better outcomes in neurocognitive tests. As expected, between-cluster differences in EFs were not attributable to differences in age or sex. Contradicting our hypothesis, but in accordance with our findings on lacking associations between EF performance and clinical characteristics as well as the results by Renwick et al. [32], clusters did not differ in BMI, level of depression, and eating disorder psychopathology. However, it remains to be studied whether EF cluster membership is connected to other clinically relevant characteristics not assessed in this study. Until recently, latent class and profile analyses in the general population and patients receiving obesity surgery only identified meaningful clusters based on self-reported self-regulation and emotion regulation, which were associated with eating disorder and general psychopathology [68,69], leaving open the question of whether objectively measured EFs are suited to reasonably depict within-heterogeneity in diverse samples.

One strength of this study is the systematic analysis of group differences in EFs regarding weight status, level of depression, and eating disorder psychopathology within a mixed-sex, treatment-seeking sample with BED. BED was interview-diagnosed according to DSM-5 criteria [1]. Clinical variables were assessed with well-established questionnaires and weight status was objectively measured. Using age-specific norm data to rank neurocognitive performance minimized potential effects of age on EFs. The inclusion of a broad spectrum of general as well as food-specific neurocognitive tasks enabled a differentiated view on EFs in individuals with BED. A major limitation of this study is that within-sample groups were formed after data acquisition instead of systematically recruiting participants according to clinical parameters. As a consequence, group sizes varied with preponderance of higher weight status and psychopathological affection. Al-though the a priori defined sample size was adequately met in general, missing data were unexpectedly high for some measures resulting in insufficient power for multivariate analyses on general EFs. At the same time, a larger sample size would have enabled using a model-based method that provides probabilities of cluster membership and does not require determining the number of clusters in advance, such as latent profile analysis [70]. Although norm-based EF ranking was provided, this study did not include a weight-matched control group that would have allowed for a weight-controlled analysis of the role of BED in EF deficits. Note also that this study used cross-sectional data on EFs in BED and can, therefore, not provide information on causal associations regarding the development and maintenance of the disorder.

## 5. Conclusions

Overall, our results show that despite some heterogeneity in EFs among adults with BED, there are hardly any global impairments present with EF performance majorly falling within the normative range. The fact that EFs were primarily independent of participants’ weight status, level of depression, and eating disorder psychopathology indicated that these clinical parameters are not suited as proxies for patients’ neurocognitive impairments. However, clinically, the individual evaluation of EFs in patients with BED is still recommended considering that up to 52% of the sample scored below average in specific EF domains. Given that BED-specific treatment, such as CBT as the first-line treatment recommendation [71], requires patients’ planning and organizational abilities [72], low levels in these EF domains may prevent optimal treatment benefit or may overburden patients, although evidence is low in this context. At the same time, this study provided evidence for distinct EF profiles, whose clinical relevance has to be further determined in future studies as identifying specific EF profiles may ultimately help clinicians to tailor treatment approaches aiming at EF improvement. To date, evidence on neurocognitive trainings for BED and obesity is equivocal [73] and studies on predictors for treatment outcomes are lacking. Randomized controlled trials on neurocognitive training in adults with BED and/or obesity did not yield any significant post-treatment differences compared to control groups in the reduction of binge-eating episodes [74,75] and in weight loss [75,76], but in general eating disorder psychopathology only [75]. Evaluating the individual need for these treatments through EF profiling could pave the way for more successful outcomes. Participants with BED were likely to perform within the normal range in computerized neuropsychological tests usually designed to quantify cognitive deficits in patients with brain damage. Scientifically, these findings highlight the need for methodological refinements of assessments in order to more precisely depict EF impairments in neurologically healthy individuals. In this context, and given the fact that the educational level differed between low and average to high EF performance clusters, patients’ individual socioeconomic background may play an important role for neuropsychological testing in BED. The relevance of EFs in BED etiology should then be further systematically and critically evaluated in longitudinal large-scale studies.

## Figures and Tables

**Table 1 brainsci-12-00006-t001:** EFs in the total sample based on normative data.

Measures	*n*	*M*	*SD*	Range	*n* (%) with *T* Score < 40/≥ 60
Cards and Lottery Task					
NAD *T* score	72	40.30	14.07	19–70	35 (48.6)/5 (6.9)
Stop-Signal Task					
SSRT *T* score	70	45.47	6.81	31–64	10 (14.3)/2 (2.9)
Go/No-Go Task					
Commission errors *T* score	76	44.57	7.97	27–68	20 (26.3)/4 (5.3)
N-Back Verbal					
Correct responses *T* score	76	47.42	8.61	27–62	12 (15.8)/12 (15.8)
Trail Making Test					
Completion time *T* score	75	49.76	8.64	29–71	8 (10.7)/10 (13.3)
Tower of London					
Planning score *T* score	70	48.97	9.59	29–80	14 (20.0)/7 (10.0)
WAFA intrinsic visual					
Mean RT *T* score	75	39.17	6.46	20–58	39 (52.0)/0 (0.0)

Note. NAD = number of advantageous decisions; RT = reaction time; SSRT = stop-signal reaction time; WAFA = Perception and Attention Functions Battery Alertness.

**Table 2 brainsci-12-00006-t002:** Means, standard deviations, and multivariate and one-way analyses of variance in measures of food bias by weight status.

Measures	OW + OB Class 1			OB Class 2			OB Class 3						
	*n*	*M*	*SD*	*n*	*M*	*SD*	*n*	*M*	*SD*	*F*	*df*	*p*	η^2^ or η_p_^2^
Food-specific EFs										3.486	4, 112	0.010	0.11
Food-specific DPT													
Food attentional bias, ms	28	−6.59 ^a^	26.16	19	13.84 ^b^	26.53	26	−12.19 ^a^	28.66	5.377	2, 70	0.007	0.13
Food-specific AAT													
Approach bias high-caloric, ms	24	3.33	82.75	14	18.64	154.30	24	−10.35	102.36	0.315	2, 59	0.731	0.01

Note. AAT = Approach Avoidance Task; DPT = Dot Probe Task; EFs = executive functions; OB = obesity; OW = overweight. ^a, b^ Means with different superscripts differ at the *p* < 0.05 level in Bonferroni-corrected post-hoc comparisons.

**Table 3 brainsci-12-00006-t003:** Mean *T* scores, standard deviations, and one-way analyses of variance for cluster indicators covering the main components of general executive functions and food bias by cluster membership.

Measures	Total *n*	Cluster 1 “Low General EFs”			Cluster 2 “Low General and Food-Biased EFs”			Cluster 3 “High EFs”			Cluster 4 “Average EFs”						
		*n*	*M*	*SD*	*n*	*M*	*SD*	*n*	*M*	*SD*	*n*	*M*	*SD*	*F*	*df*	*p*	η^2^
Cards and Lottery Task																	
NAD	72	29	29.14 ^a^	7.16	11	31.39 ^a^	9.81	19	51.27 ^b^	6.09	13	56.67 ^b^	7.84	61.144	3, 68	<0.001	0.73
Stop-Signal Task																	
SSRT	70	28	45.21 ^a, b^	6.03	10	44.30 ^a, b^	7.26	20	49.40 ^a^	6.87	12	40.50 ^b^	4.54	5.293	3, 66	0.002	0.19
Go/No-Go Task																	
Commission errors	76	29	42.90	7.51	12	45.00	7.25	21	45.33	9.36	14	46.50	7.35	0.769	3, 72	0.515	0.03
N-Back Verbal																	
Correct responses	76	29	45.52 ^a, b^	7.58	12	42.25 ^a^	9.01	21	50.71 ^b^	9.57	14	50.86 ^a, b^	5.60	4.146	3, 72	0.009	0.15
Trail Making Test																	
Completion time	75	28	49.93 ^a^	8.44	12	49.00 ^a, b^	8.52	21	54.67 ^a^	7.05	14	42.71 ^b^	6.83	6.620	3, 71	<0.001	0.22
Tower of London																	
Planning score	70	26	46.27 ^a^	9.59	12	45.25 ^a^	7.02	20	56.05 ^b^	7.49	12	46.75 ^a^	9.66	6.319	3, 66	<0.001	0.22
WAFA intrinsic visual																	
Mean RT	75	28	38.89	7.38	12	39.08	5.02	21	39.57	4.93	14	39.21	8.10	0.043	3, 71	0.988	0.00
Food-specific DPT																	
Food attentional bias	73	28	53.91 ^a^	9.19	9	36.12 ^b^	7.04	21	54.02 ^a^	5.85	15	45.39 ^c^	8.53	14.802	3, 69	<0.001	0.39
Food-specific AAT																	
Approach bias high-caloric	62	23	55.43 ^a^	9.67	8	36.69 ^b^	3.93	18	48.38 ^a^	6.17	13	50.83 ^a^	9.72	10.545	3, 58	<0.001	0.35

Note. AAT = Approach Avoidance Task; DPT = Dot Probe Task; EFs = executive functions; NAD = number of advantageous decisions; RT = reaction time; SSRT = stop-signal reaction time; WAFA = Perception and Attention Functions Battery Alertness. Measures of general EFs are *T* scores derived from representative samples [35] and published data [36]. Measures of food bias were converted into *T* scores to ensure comparability. Lower values indicate worse EFs/higher food bias. ^a, b, c^ Means with different superscripts differ at the *p* < 0.05 level in Bonferroni-corrected post-hoc comparisons.

**Table 4 brainsci-12-00006-t004:** Demographic and clinical characteristics of clusters according to their executive functions profile.

Measures	Cluster 1 “Low General EFs”		Cluster 2 “Low General and Food-Biased EFs”		Cluster 3 “High EFs”		Cluster 4 “Average EFs”					
	*M* or *n*	*SD* or %	*M* or *n*	*SD* or %	*M* or *n*	*SD* or %	*M* or *n*	*SD* or %	*F* or χ²	*df*	*p*	η^2^ or Cramér’s *V*
Demographics												
Sex, female	23	79.31	11	91.67	13	61.90	13	86.67			0.211	
Age, years	51.54	11.01	45.76	11.34	42.26	13.19	42.27	15.31	2.914	3, 73	0.040 ^†^	0.11
Educational level, ≥ 12 years	9 ^a^	31.03	2 ^a^	16.67	14 ^b^	66.67	12 ^b^	80.00	17.148	3	<0.001	0.47
Clinical parameters												
BMI, kg/m²	36.30	4.91	38.14	3.25	37.46	5.41	37.00	5.39	0.463	3, 73	0.709	0.02
EDE-Q global score (0–6)	2.90	1.13	2.59	1.22	2.59	1.01	3.21	0.92	1.151	3, 70	0.335	0.05
PHQ-9 sum score (0–27)	9.66	5.15	8.36	3.61	7.47	4.72	8.67	3.13	0.938	3, 70	0.427	0.04

Note. BMI = body mass index; EDE-Q = Eating Disorder Examination-Questionnaire; EFs = executive functions; PHQ-9 = Patient Health Questionnaire depression module. ^†^ Group means did not differ at the *p* < 0.05 level in Bonferroni-corrected post-hoc comparisons. ^a^ Count significantly lower than expected at the *p* < 0.05 level. ^b^ Count significantly higher than expected at the *p* < 0.05 level.

## Data Availability

The data are not publicly available.

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
