# Peer review of "Executive Functions of Adults with Binge-Eating Disorder: The Role of Weight Status and Psychopathology"

_brainsci, 2021, doi:10.3390/brainsci12010006_

Round 1

Reviewer 1 Report

-

Reviewer 2 Report

In this work, Busch et al. tackle a series of limitations found within the BED literature with regards to EF deficits. Such limitations include design/methodological (small sample sizes, paradigms unrelated to food, lacking norms in in-house measurements) conceptual concerns (body-weight status, depression symptomatology, or BED severity). To this end, first, authors compared BED performance on both global and food-specific EF with population-based norms, expecting to find below-average performance in BED. Second, they tested the effects of weight status, symptoms of depression and BED severity on EF performance. Authors anticipated all three to be negatively related to EF performance. Third and last, they performed a cluster analysis to classify individuals into subtypes based on their performance on three EF domains (inhibition, working memory and cognitive flexibility) and 2 measures of food bias (attentional and approach toward hi-cal foods). In accordance with the literature, authors expected a 3-cluster solution with one cluster showing general below average EF performance, another cluster showing above-average EF performance and a third and last cluster presenting food-specific impairments. In addition, the above-average cluster will present lower weight status, depression symptoms and BED severity.

Contrary to what authors hypothesized, (1) most BED patients did not score below average (± 10 T-score). Similarly, differences in weight status, symptoms of depression or BED severity did not affect EF performance but only attentional bias towards food in a non-linear fashion (i.e., obesity type 2 performing worse than overweight/obesity type 1 and obesity type 3). In addition, results from the cluster analysis were not the ones authors anticipated – the analysis did not return a highly-functioning subtype but two with different strengths/limitations, and none showed lower weight status, depression symptoms nor BED severity.

Authors conclude that despite in their analyses neither weight status, depression symptoms or BED severity seemed to have had the clinically negative impact on EF performance, the BED group showed below-average performance, which could affect adherence to and outcomes from CBT. Subtyping different phenotypes of BED could also revest some value in those interested in more tailored-based approaches.

In general, I enjoyed reading this work. Transitions from one paragraph to the other are smooth and logical. The research topic is interesting and authors’ intentions in tackling heterogeneity in BED literature is remarkable. I do, however, have some concerns about some decisions as to the measures and analyses involved that I will cover in more detail in the next paragraph. There are other minor comments in the shape of suggestions to make some parts of the manuscript easier for the reader. I leave to the authors whether these make it to the final version of the work.

*Major* comments:

Methods

  • Discrete independent variables. I don’t follow the rationale for transforming continuous variables such as BMI, PHQ-9 or EDE-Q scores into discrete variables (e.g., no depression, mild depression, severe depression). Could the authors comment on what the advantage of doing this is? Do results change if continuous variables are used instead in the second analysis?

  • Go/no-Go. I appreciate the detail in which authors describe all their measures. However, and regarding the Go/no-Go paradigm, what was the ratio of Go trials vs no-Go trials (i.e., triangles vs circles)? This detail should be clearly stated.

  • Outliers’ treatment. I was not very much concerned about this given the low number of excluded participants (1 to 3). However, in line 270 (page 6) it seems the MANOVAs were run on 57 and 60 subjects, numbers that were very much below expected after removing the few mentioned outliers (total sample 77). Could the authors please provide a diagram or flow chart explaining how many participants made it to each of the three analyses conducted? (BED vs norms, BED vs covariates of interest, subtyping).

  •  In line 283 (page 6) authors say that to reduce complexity only three core EF domains will be used along with two food-specific tests to determine subtypes of interest. After introducing a *very generous* battery of cognitive measurements, it seems atypical to have had left planning, decision-making, and alertness out of this clustering procedure.

Authors expected below-average, above-average and food-specific impairments clusters. I can’t think of a reason why including high-order EFs and alertness would possibly ‘hurt’ the interpretation of the clusters. Additionally, if data dimensionality was the issue (which it shouldn’t be, clustering methods can handle several variables) authors could have tried dimensionality reduction methods (e.g., PCA) as a front-end step.

Also, a very important point, how many cluster solutions were tested? In unsupervised learning methods like this, it is the user that specifies how many clusters they want (i.e., default usually is 2 to 6).

Results

  • Norm-based tests (hypothesis 1). It is not clear what is the expected result. I understand that authors are counting how many BED participants are below a T-score of 40 by looking at Table 1 on page 7 and that if the average of the BED is lower than 40 then the conclusion is BED patients show deficits. Furthermore, lacking a control group is one major limitation that drastically restricts the type of analyses to conduct. This has not been acknowledged as a limitation (and it should be).

  • Why years of education (or educational level) was not included as a nuisance covariate in analyses 2 and 3? I understand that in the first analysis the distribution of educational level was equally distributed, but further grouping (analysis 2) and subtyping (analysis 3) can be problematic. Imagine the obesity type 3 group has a greater number of higher educated subjects explaining the lack of differences with obesity type 1 and overweight in food attentional bias? In this line, while age is somehow ‘controlled’ because of the use of age-specific T scores, why is sex not also further controlled in the MANOVAs with education? (note: in the clustering analysis sex does not differ between the groups, so there is no need to include sex in that analysis)

  • Cluster solution. Could the authors provide a figure with the Ward’s coefficient on the Y-axis and the number of cluster solutions tested in the X-axis? This would add robustness to their decision of using 3 clusters.

Also related to this, why performance on planning, decision-making and alertness was not compared between the 3 clusters?

*Minor* comments:

           In general, the manuscript would benefit from additional visual support of the analyses conducted.

Reviewer 3 Report

I appreciate the opportunity to review this article, and I congratulate the authors for the topic and the methodology developed. I just have a few small suggestions:

- Indicate why the PHQ-Q and EDE-Q questionnaires were chosen, and not other more common ones for the evaluation of depression such as the BDI or the EAT for eating disorders.
- Similarly, justify why the different tasks were chosen to evaluate executive functions and not others.
- Assess the clinical implications of the results in the discussion-conclusions section.

Round 2

Reviewer 2 Report

Thank you for addressing all my comments. I don’t have any further suggestions. Congratulations on the paper.